# Metabolomics Reveals Heterogeneity in the Chemical Composition of Green and White Spears of Asparagus (*A. officinalis*)

**DOI:** 10.3390/metabo11100708

**Published:** 2021-10-16

**Authors:** Eirini Pegiou, Qingrui Zhu, Paraskevas Pegios, Ric C. H. De Vos, Roland Mumm, Robert D. Hall

**Affiliations:** 1Laboratory of Plant Physiology, Wageningen University and Research, 6700AA Wageningen, The Netherlands; eirini.pegiou@wur.nl (E.P.); qingruizhu@outlook.com (Q.Z.); 2Laboratory of Food Chemistry, Wageningen University and Research, 6700AA Wageningen, The Netherlands; 3DTU Compute, Technical University of Denmark, 2800 Lyngby, Denmark; ppegiosk@gmail.com; 4Business Unit Bioscience, Wageningen Plant Research, Wageningen University and Research, 6700AA Wageningen, The Netherlands; ric.devos@wur.nl (R.C.H.D.V.); roland.mumm@wur.nl (R.M.)

**Keywords:** *Asparagus officinalis*, asparagus metabolome, GC-MS, LC-MS, crop metabolomics, spatial metabolomics

## Abstract

Green and white asparagus are quite different crops but can be harvested from the same plant. They have distinct morphological differences due to their mode of cultivation and they are characterised by having contrasting appearance and flavour. Significant chemical differences are therefore expected. Spears from three varieties of both green and white forms, harvested in two consecutive seasons were analysed using headspace GC-MS and LC-MS with an untargeted metabolomic workflow. Mainly C_5_ and C_8_ alcohols and aldehydes, and phenolic compounds were more abundant in green spears, whereas benzenoids, monoterpenes, unsaturated aldehydes and steroidal saponins were more abundant in white ones. Previously reported key asparagus volatiles and non-volatiles were detected at similar or not significantly different levels in the two asparagus types. Spatial metabolomics revealed also that many volatiles with known positive aroma attributes were significantly more abundant in the upper parts of the spears and showed a decreasing trend towards the base. These findings provide valuable insights into the metabolome of raw asparagus, the contrasts between green and white spears as well as the different chemical distributions along the stem.

## 1. Introduction

Asparagus (*Asparagus officinalis*) is a perennial crop, where the spears (shoots) can be eaten as a nutritious vegetable. It has two main types: green and white which can be derived from the same variety, depending on the cultivation method. Green asparagus is harvested above-ground, while white asparagus grows fully immersed in soil wherein it is also harvested [1]. The key difference in cultivation relates to growth with or without sunlight. Green asparagus is, by far, the most widely grown whereas white asparagus is more country-specific. In Europe, for example, green asparagus is more commonly grown and consumed in, e.g., the UK and Spain while white asparagus is more popular in, e.g., Germany and The Netherlands. In some countries, consumers speak about “asparagus” without distinguishing the type as they are unaware of the white alternative [1].

For cultivating asparagus, the crowns should be planted such that they are ca. 40 cm deep. For green asparagus, once the shoots start to grow, they emerge above the soil, are exposed to sunlight, and elongate further. For white asparagus, ridges are made using an extra ca. 20 cm of topsoil and the shoots are harvested while still beneath the soil as they must remain in the dark [1]. Exposure to light not only induces the synthesis of pigments such as chlorophyll and anthocyanins, but also strongly determines spear morphology (Figure 1a). Green asparagus spears bear more prominent cladodes (leaf-like organs) (Figure 1b). Furthermore, white spears are generally significantly thicker, as in darkness, cells from the primary thickening meristem proliferate both periclinally and anticlinally [2,3].

The composition of secondary metabolites in plants is tissue-specific and is influenced by developmental stage and genetic/environmental perturbation [4]. Many different pathways are involved in secondary metabolite biosynthesis, including the phenylpropanoid, methyl-erythritol-4-phosphate (MEP) and mevalonate (MVA) pathways, which lead to the formation of many different volatile and non-volatile (poly)phenolic and terpenoid compounds [4]. Volatile organic compounds (VOCs) formulate the aroma profile and, in addition to the aforementioned pathways, many other aroma compounds are formed by the conversion/degradation of fatty acids and amino acids into straight and branched medium-chain aldehydes, alcohols and esters [4,5,6], as well as heterocyclic compounds such as methoxypyrazines [7].

Aroma is one of the two key determinants of asparagus flavour [8,9,10,11]. Thanks to the studies of Hoberg, Ulrich et al. on the aroma profile of cooked white asparagus [10,11,12,13], we can now refer to specific VOCs (e.g., dimethyl sulfide, 2-methoxy-3-isopropyl pyrazine, methional) as being the “key odorants” also currently used as industrial references for flavour quality. Remarkably little research has been done on the metabolite profile of the raw vegetable. Such investigations are needed to provide the fundamental chemical basis to understanding the physiology and chemistry of the crop and to explore precursor pathways involved in the formation of flavour also after food preparation. It is hypothesised that the flavour chemistry of green and white asparagus should have many commonalities, as they both have a recognisable asparagus taste, while contrasting compounds will be the basis of the nuances in green/white flavour.

Several studies on green or white asparagus have focused on the chemical composition of the crop, including, e.g., minerals [14], flavonoids [15,16] and saponins [17,18]. However, most studies only involve green asparagus although sometimes the type is not actually mentioned. A few studies have compared the chemical composition of both types, covering mineral composition [19], sensory characteristics [20], antioxidant capacity [21], and the quantification of specific chemicals such as the flavonoid rutin and the saponin protodioscin [22]. These last two studies also compared different regions within the spears. Nevertheless, conclusions were limited as this work was performed using materials from different cultivation locations or used shop-bought materials of unknown varietal origin or cultivation history.

During/after harvest much asparagus material is discarded [1]. There is now growing interest to exploit this material to extract flavour compounds for the food industry, aiming for a more sustainable food production system. Most of this waste comprises stem bases [23]. The study described here was performed with two main goals: to exploit untargeted metabolomics to investigate the chemical differences between green and white spears and to relate chemical differences to the contrasts in spear development and growth regime. We also performed spatial metabolomics to localise sensory-relevant metabolites within the spear in the context of exploiting waste materials in a circular bioeconomy strategy. The strength of our conclusions has benefitted greatly from the fact that we were able to access all materials, both green and white, of three known, validated varieties all of which were grown at a single commercial field location and all were harvested at the same specific times.

## 2. Results

### 2.1. Untargeted Metabolomics of Green and White Asparagus Spears

Three asparagus varieties in both green and white form were harvested and analysed per platform in a single randomised sample sequence, for 2 successive seasons (2019 and 2020). Both GC-MS and LC-MS raw data were processed in an untargeted manner based on the dedicated MetAlign-MSClust workflow. A number of technical replicates of a mixed sample were also analysed in each series to assess technical reproducibility (Quality Controls: QCs). The average variation in metabolite abundance within the QCs (RSD) was <20%, which is within acceptable limits for untargeted metabolomics studies.

Clear quantitative and qualitative differences were revealed with a first inspection of the raw GC-MS and LC-MS data plotted as total ion current (TIC) and base peak chromatograms, respectively (Figure 2).

#### 2.1.1. Volatile Secondary Metabolites in Green and White Asparagus

In total, 104 VOCs were detected in the headspace of green and white asparagus samples from the two harvest years, of which 88 were identified (including 61 common metabolites from 2019 and 2020). Of these 88 VOCs, 33 were unambiguously identified and 55 were tentatively annotated. The asparagus volatile profile mainly consisted of alcohols, aldehydes, esters, ketones and short-chain fatty acids with 2–13 C atoms (C_2_–C_13_). Furthermore, 9 monoterpenoids and 10 benzene derivatives were tentatively identified. Other identified VOCs included 5 sulfur-containing compounds (S-compounds), namely methanethiol, dimethyl sulfide, dimethyl disulfide, dimethyl trisulfide and methional, together with 4 furans and 2 methoxypyrazines (Appendix A). The abundances of >60% of the detected VOCs in both years were significantly different between the two asparagus types (Appendix A). The VOCs detected in both years as being discriminatory for the two types are summarised in Table 1a.

#### 2.1.2. Non-Volatile Secondary Metabolites in Green and White Asparagus

A total of 442 semi-polar compounds were detected in the asparagus spears. Based on the reconstructed (in-source) mass spectra provided by the MSClust tool, 76 compounds could be assigned with a monoisotopic mass and 56 could be putatively identified. Mainly flavonoids, other phenolic compounds, triterpenoid glycosides (saponins) and S-compounds were among the identified non-volatile secondary metabolites. At the beginning of the chromatograms, additional polar compounds could be detected, and within these, arginine, aspartic acid, citric acid, sucrose, raffinose and uridine diphosphate glucose were tentatively identified (Appendix A). However, due to the high possibility of ion suppression effects in this crowded part of the chromatogram, these polar primary metabolites were not further examined in this study.

Not all non-volatile components were detected both in green and white asparagus. Both in 2019 and 2020, >50% of the detected compounds were discriminatory for the two asparagus types (Appendix A). For those overlapping in both seasons, the ones that could be assigned with a monoisotopic mass and identified are shown in Table 1b.

### 2.2. Comparison of the Chemical Composition of Green and White Asparagus in 2019

Following the processing of the GC-MS and LC-MS data, a principal component analysis (PCA) was performed separately per platform for the green and white asparagus data. In both cases, the QC samples are closely grouped and located in the middle of the plots, showing good technical reproducibility of both analytical platforms (Figure 3). Clearly, both the volatile and the non-volatile profiles of the two types are distinct, as in both datasets the green and white asparagus spears are separated along the first principal component (PC1), which explained 57.3% and 50.6% of the variation between the samples, respectively (Figure 3).

For both platforms, most (>50%) compounds were detected at significantly different levels (adjusted *p* value ≤ 0.05) between the two types, confirming the PCA findings (Appendix A). In the case of VOCs, some alcohols and aldehydes, as well as a few short chain fatty acids and S-compounds were more abundant in the green type, while most benzene derivatives, monoterpenoids and some aldehydes were more abundant in the white type (Appendix A and Appendix A). Concerning the non-volatiles, the clearest distinction between the two asparagus types was in the composition of phenolics (more abundant in green) and saponins (more abundant in white) (Appendix A and Appendix A).

In both PCA plots, PC2 depicts an additional ca. 10% of the overall variation (Figure 3). Although the white asparagus varieties are all closely grouped for both platforms, the green varieties are separated across PC2 (Figure 3). Specifically, with regard to the volatile profiles the green varieties are all separate (Figure 3a), and for the non-volatile profiles, especially the green Avalim had a distinct profile (Figure 3b). The abundances of 11 VOCs and ca. 16% of the detected non-volatiles were found to significantly vary between the three green varieties (Appendix A).

In 2019, for practical reasons not all green varieties could be harvested at exactly the same time; Gijnlim and Grolim were harvested in calendar week 21 and the green Avalim was harvested in calendar week 23. Given this difference in harvest-day between the green varieties, their metabolite profiles (volatiles and non-volatiles) were compared to determine if there was a potential harvest-day effect. The levels of the same compounds varying between the green varieties were also found to vary between the green asparagus harvested on the two different days in 2019 indeed suggesting a harvest-day effect. These compounds were ethanol, acetic acid, ethyl acetate, (*E*,*E*)-2,4-heptadienal, 2-methoxy-3-isopropylpyrazine, acetaldehyde, chlorogenic acid, and other detected flavonoid glucosides (Appendix A). A more in-depth analysis was therefore performed in the follow-up validation study in 2020.

### 2.3. Comparison of the Chemical Composition of Green and White Asparagus in 2020

To validate and follow-up on the results from the first year’s analyses, in 2020, the same green and white varieties were harvested from the same field as in 2019. All varieties were harvested twice, with three weeks in between, to investigate potential genotype and harvest time-point differences. The sample preparation, the analyses, the examination and processing of the raw data were all carried out in the same way as in 2019.

PCA was performed separately on the complete 2020 datasets, both for the GC-MS and the LC-MS data. The QCs again indicate good reproducibility of the analyses (average RSD based on compounds present in QCs < 20%) and the green and white asparagus samples are again separated along PC1 (Figure 4). Further statistical analyses showed the discriminatory metabolites for the two asparagus types of the 2020 season which corresponded to 64% and 75% of the detected VOCs and non-volatiles, respectively (Appendix A). In agreement with the 2019 data, aldehydes and alcohols were highly abundant in the green varieties while white spears were characterised with high levels of a few other aldehydes. The differences in the levels of all benzenoids and monoterpenoids between green and white asparagus were less distinct based on the differential PC1 values (Appendix A). With regard to the non-volatiles, the metabolites separating the green and white spears comprised polyphenols and saponins, respectively (Appendix A), confirming the findings of 2019. Additionally, PCA was performed on the combined 2019 and 2020 GC-MS data on the 61 identified volatiles detected in both seasons verifying the distinct differences in the volatile composition of green and white spears (Appendix A).

The separation of the green varieties based on their volatile and non-volatile profiles along PC2 that was observed in 2019 data, was less evident in the 2020 data (Figure 4). Further analyses comparing the profiles of the three varieties were performed, separately for the green and white asparagus and only 6 volatiles and 0 non-volatiles were found at significantly different levels (adjusted *p* value ≤ 0.05) between the three varieties in either the green or white type (Figure A1 and Figure A2, Appendix A).

Within each asparagus type ca. 25% of the total variation in the profiles of volatiles and non-volatiles was caused by the harvest time-point, which also appeared to be type dependent (Figure A1 and Figure A2 in Appendix B). The compounds significantly changing in calendar week 23 compared to week 20 are 2-methylfuran, toluene, 2-methoxy-3-isopropylpyrazine, 2-methoxy-3-isobutylpyrazine and methanethiol for the white spears and chlorogenic acid, one asparagusic acid ester isomer, heptanal, 3-methylthiopropanal (methional), nonanal and octanal for the green spears (Appendix A).

### 2.4. Non-Discriminatory Compounds in Green and White Asparagus

Metabolites that were detected at similar (FC < 2) or not significantly different levels (*p* value > 0.05) between green and white asparagus are also of interest. Such metabolites can be expected to be involved in general asparagus chemistry and contribute to the typical asparagus flavour characteristics, common to both types.

For both years, the metabolites with less than a two-fold difference between the two types were 2-heptanone, 2-methyl-3-buten-2-ol, 2-methylfuran, (*E*)-2-hexenal, methanethiol, (*E*)-2-octenal, 1-octen-3-one, hexanal, heptanal, (*E*)-2-heptenal, 2-pentylfuran, pentanal, dimethyl sulfide, dimethyl disulfide, 2-methoxy-3-isopropylpyrazine, asparagusic acid and its ester isomers (Appendix A). Moreover, the two asparagus types did not significantly differ with regard to 3-methyl-1-butanol, (*E*)-2-hexen-1-ol, 1-octen-3-ol, dimethyl trisulfide, 2-methoxy-3-isobutylpyrazine and asparaptine (Appendix A).

### 2.5. Spatial Metabolomics of the Asparagus Spear

Throughout the asparagus season, large waste-streams are generated which do not just consist of complete spears and indeed mainly comprises stem bases. It was hypothesised that there is a difference regarding the volatile composition along the spear. To test this hypothesis, a GC-MS analysis was performed to profile the secondary VOCs of 5 (for green) to 6 (for white) different spear sections taken from top to base.

The majority of the detected VOCs showed compositional heterogeneity along the asparagus spear for both the green and white types. A number of VOCs (20 in white, 11 in green spears) were more abundant in the upper parts with a decreasing trend towards the base, as exemplified by, e.g., 1-penten-3-ol, 1-pentanol and 1-octen-3-ol. The opposite pattern was also found, as in the case of 4-methyl-3-pentanone. Interestingly, a few compounds, such as 2-methoxy-3-isopropyl pyrazine and 2-pentylfuran, showed contrasting gradients between the two asparagus types (Figure 5).

## 3. Discussion

This comprehensive (untargeted) metabolomics research explored the complexity of green and white asparagus chemistry, how this is environmentally, developmentally, spatially and genetically influenced and attempts were also made to link this to differences in asparagus flavour and sensory relevance.

### 3.1. Asparagus Metabolome and Differences in Chemical Compositions of Green and White Spears

This study is the first to compare green and white asparagus volatile profiles in a holistic way revealing greater detail of their biochemical composition. Some of the identified aldehydes have been detected earlier in asparagus, such as pentanal, hexanal, (*E*)-2-hexenal, heptanal, and octanal [12,25,26]. Other aldehydes detected in this study are reported in raw asparagus for the first time. Some of these, such as phenylacetaldehyde (benzeneacetaldehyde) and 3-methylbutanal are known vegetable VOCs, and their aroma attributes are green, fresh, floral, and fruity [27](www.foodb.ca, accessed on 13 September 2021). Among the unambiguously identified alcohols and ketones, 1-octen-3-ol, 1-penten-3-ol, 1-pentanol, and 2-butanone have been reported in (cooked) asparagus [8,10,12,13,25]. Furthermore, most of the monoterpenes and benzenoids, which are here reported for the first time in asparagus, are also already-known vegetable/fruit VOCs. For instance, styrene and limonene have been detected in blackcurrant, cauliflower, apple, cabbage, celery, carrot, etc., and their aroma attributes contain several fruity, fresh, and floral notes [27].

Interestingly, in this study, a number of VOCs that have previously been reported in other asparagus materials (e.g., cooked white spears) were not detected. A few examples are 2,3-butanedione, 2,3-pentanedione, or 2,5-dimethylpyrazine [1]. These are volatile compounds usually formed via heat-induced reactions [28] and so this might explain their absence in the raw spears. With regard to the non-volatiles, a large number of the identified compounds have been previously reported in asparagus. These include compounds such as rutin [22], 3-O-feruloylquinic acid [29], asparaptine [30], protodioscin [31], as well as asparagusic acid [9] and its ester isomers [32]. However, it should be emphasised that here, when using a non-targeted LC-MS approach without (semi)automated MS/MS spectral analysis, the majority of the metabolites detected could not readily be identified, indicating that there is a wealth of chemical information still waiting to be discovered.

One highlight of the study presented here is the list of discriminatory secondary metabolites for green and white asparagus, which was validated after analyses in two consecutive seasons (Table 1). In particular, some of the compounds that characterised the white spears were 1,2-dimethoxybenzene, (*E*,*E*)-2,4-decadienal, shatavarin and protodioscin, while green spears were distinguished by significantly higher levels of 1-penten-3-ol, 3-pentanone, octanal, rutin and most of the identified phenolics (Table 1, Appendix A, Appendix A).

These differences might primarily arise due to their growing environment and the contrasting developmental processes which the two types follow. Green asparagus grows above the ground, while white asparagus stays below ground and often under impermeable plastic foil to avoid any exposure to sunlight. The distinct chemical differences between the two asparagus types (Figure 3 and Figure 4, Table 1) imply that biochemical routes are differentially active in green and white shoots. These contrasts may be directly connected to the light/dark conditions of growth as we are comparing photosynthetic and non-photosynthetic systems where many differences might be anticipated. However, it is perhaps surprising how many common metabolites were found within the two systems although significant quantitative differences could be observed which may or may not be linked to the photosynthetic state. Worthy of mentioning here is that the metabolomics methods applied are not suitable to analyse the specific photosynthetic (apolar) thylakoid-bound compounds in green plant tissues like chlorophylls and carotenoids.

Different environments lead to the development of different plant traits [33], and this can also be translated into variation in their metabolomes. Next to light, green asparagus is also more readily exposed to any changes in weather conditions and especially, temperature fluctuations [34]. Weather changes can result in the production and emission of C_5_–C_8_ aldehydes and alcohols [35], which here were found to be more abundant in the green spears (Table 1a and Appendix A). The biosynthesis of plant polyphenols is linked to light and especially, UV exposure, where induced accumulation of phenolic compounds has been linked to plant protection [36]. In this study, such compounds were detected only (e.g., 3-O-feruloylquinic acid), or were at considerably higher levels (e.g., rutin), in green asparagus (Table 1b and Appendix A). On the other hand, white asparagus shoots grow entirely underground which can have a certain buffering/protective capacity to the growing shoots thus limiting weather-related abiotic stress fluctuations. In this study, a few benzenoids, monoterpenoids, unsaturated aldehydes and saponins were found to be more abundant in white asparagus (Table 1 and Appendix A). Subterranean organs in plants have evolved specific protective (chemical) mechanisms to withstand attack by pathogens and microorganisms [37]. Some of these protectants might be carried over from the roots to the young shoots when still below ground. For example, higher saponin levels are typical of asparagus roots [18,38] as well as white shoots compared to green ones (Table 1). In this regard, David and Hofmann (2014) [39] and Lee et al., (2010) [22] found higher concentrations of saponins in the stem sections closer to the crown/roots of white asparagus, compared to the young asparagus tips.

This differentiation of the profiles in flavonoids (more abundant in green) and saponins (more abundant in white spears) might have been anticipated since previous studies have reported generally high flavonoid contents in photosynthetic tissues [40], and high saponin levels in plant parts growing below-ground [18,38]. Using a complementary approach, Yi et al., (2019) performed de novo transcriptome sequencing in both green and white types of a commercial variety (Atlas) and found significant differences between the two asparagus types in that 21% of the genes involved in the biosynthesis of the flavonoid rutin were upregulated in green and 4% of the genes involved in the biosynthesis of the saponin protodioscin were upregulated in white asparagus [41]. These results followed and confirmed other studies, which focused on quantifying these two compounds in green and white asparagus, as well as other flavonoids and saponins [22,32,39,42,43], suggesting that indeed metabolic pathways are differently active in green and white asparagus shoots.

The distinct differences in the metabolome of green and white asparagus also likely contribute to the distinct green and white flavour profiles, respectively. Cuppett et al., (1997) performed a sensory analysis on raw materials, showing that the main flavour variations were green, grassy and bitter notes for the green asparagus, and sweet corn, potato and buttery notes for the white type [20]. This flavour variation can at least partially be explained by the specific VOCs that were shown in this study to be predominantly abundant in each asparagus type (Table 1 and Appendix A).

### 3.2. Non-Discriminatory Asparagus Compounds in Green and White Spears

Green and white asparagus spears are different but they also have many commonalities and compounds that did not significantly differ between the two types (e.g., 1-octen-3-ol, dimethyl trisulfide, 2-methoxy-3-isobutylpyrazine and asparaptine) (Appendix A) can be proposed as “typical” asparagus compounds, which may contribute to the general asparagus flavour characteristics, common to both types. In addition, the known cooked white asparagus odorants dimethyl sulfide, methional (3-methylthiopropanal), 2-methoxy-3-isopropyl pyrazine, [9,10], were also detected here in raw asparagus in both green and white types. These commonalities may emphasise the potential general importance of these sensory-relevant compounds to the overall asparagus sensory experience, as they are present in both types. Concerning the metabolites with FC < 2 (e.g., 2-methylfuran, (*E*)-2-hexenal, methanethiol and (*E*)-2-octenal), despite how small the differences between the two types, these may still have an impact on flavour perception [44]. Future investigation and focus on such compounds are of relevance for confirming the nature of any flavour deviations.

### 3.3. Variety and Harvest Time-Point Effects on the Asparagus Metabolome

Concerning genetic background, relatively minor differences were observed especially for the white asparagus type (Figure A1 and Figure A2, Appendix A). This might imply that fewer chemical differences mean similar taste and aroma profiles, but this still needs to be tested. For the green type of the varieties, there did appear to be more varietal discrimination, especially in the 2019 season, which might also be assigned to the impact of the different harvest days. Initially, as harvest day differed by just a few weeks, this effect was not anticipated but to investigate, in 2020, the validation study was specifically designed to include two harvest time-points with three weeks in between. The set-up of this 2020 study was such that the only source of variation was the cultivation method (green or white) or the genetic background, at each harvest time-point. Again, no significant variation between the three varieties was found, in either green or white type, thus confirming the findings from 2019. This apparent lack of genotypic influence, as compared to spear type and harvest period, is perhaps surprising but may be related to all three varieties having been generated by the same breeding company. It is perhaps also worthy of mentioning here that while Grolim is usually only used to produce white asparagus, while the other two varieties (Avalim and Gijnlim) are considered suitable as both green and white crops, this did not appear to be reflected in the biochemical profiles.

In the 2020 data, despite just a three week difference between harvests, clear harvest-day effects were observed in the metabolomes, which interestingly appeared to be type dependent (Appendix A). Importantly, these included the VOCs dimethyl disulfide, methional and 2-methoxy-3-isopropyl pyrazine. All three are known asparagus odorants [10]. Such temporal variation in the asparagus metabolome has also been observed by Creydt et al. (2018) [45], who suggested that even a one week period can have a high impact. These observations suggest that the asparagus aroma profile is dynamic and likely also (harvest) time dependent, but this would need further investigation using larger sample sets, in combination with taste trials.

For the non-volatiles, only a few compounds (27% of total detected) appeared to be dependent on harvest day, but these did not include the putatively identified polyphenols and saponins, which were most abundant in green and white asparagus, respectively (Table 1 and Appendix A). Soteriou et al., (2021) have recently shown that there was a significant increase in total phenolics and a significant decrease in total sugars towards the end of the asparagus harvest season [46]. This can be due to the fact that they sampled at longer time intervals across a 10-week season while we used only two time-points separated by 3 weeks. Future research should focus on fully defining the dynamics of variation on the asparagus metabolite profiles throughout a complete harvest season. This work is now underway.

### 3.4. Metabolite Distribution along the Asparagus Spear

Aiming to understand better the potential sensory quality of raw asparagus and whether there is a variation of metabolite composition along the shoot, different regions of the asparagus spear were analysed using GC-MS. Several VOCs were observed to have a compositional gradient from top to bottom (e.g., 1-penten-3-ol) or from bottom to top (e.g., 4-methyl-2-pentanone) of the spear (Figure 5). This observed heterogeneity in the distribution of the volatiles along the green and white spears matches with previous studies on asparagus which have focused on other types of analytes. For example, Amaro-López (1998) [14] and Makus (1994) [19] studied the distribution of important minerals in the green and white asparagus shoots, and Creydt and Fischer (2020) [47] have recently investigated the distribution of primary metabolites in white asparagus spears. Moreover, Dawid and Hofmann (2014) [39] have demonstrated a quantitative gradient of asparagus saponins along the white asparagus shoot, while Lee (2010) [22] and Wang (2003) [43] have also shown a gradient of protodioscin along asparagus spears. Together, all these complementary observations clearly indicate considerable heterogeneity in metabolite presence within asparagus spears.

Given the rapid growth of the asparagus shoots during the season, various developmental stages can be found along the spear, from the large upper meristem comprising mainly mitotic cells to the lower regions where only cell expansion is likely. Therefore, a more active metabolome might be expected in the young asparagus tips compared to the lower parts and considering their greater vulnerability to herbivore and pathogen attack, a more extensive protective chemical arsenal may have evolved. In this study, the metabolite composition in the tips was indeed generally chemically richer (Figure 5) than that of the basal regions and complements preliminary findings by Nakabayashi et al. (2021), who described the localisation of asparaptine in asparagus shoot apical meristems [48].

This observed heterogeneity in the chemical composition has particular relevance in the context of circularity and the exploitation of waste streams [49]. During the asparagus harvest season, ca. one-third of the harvested asparagus material is discarded as waste; both as imperfect whole spears as well as the trimmed-off stem bases removed to produce equally-long spears for the supermarket [1,23]. This huge volume of biomass is seen to be a potentially valuable source of asparagus aroma and flavour chemicals [23]. Considering the chemical variation observed here, it should therefore be born in mind that different waste streams may produce different quality end-products. For example, as the majority of the VOCs with fruity, fresh, bitter and floral aromas, were found to be more abundant in the upper parts of the asparagus spears, waste materials containing more of this tissue and fewer stem bases will likely yield a richer aroma profile which is also more typical of the whole spear. However, this needs further investigation and verification as not all compounds contribute equally to sensory impact hence such an analysis needs to involve both profiling of the key bioactive metabolites and proper aroma and flavour evaluation.

## 4. Materials and Methods

### 4.1. Asparagus Plant Materials

All materials used in this study originated from a single field location where the green and white crops were grown side by side. The green and white spears of three asparagus varieties were included, Avalim, Gijnlim and Grolim. All three varieties are 100% hybrid F1 plants. It is relevant to note that while both Avalim and Gijnlim are considered suitable for both green and white asparagus production, Grolim is less suited as a green crop because it can have a tendency to give more open heads under sub-optimal temperatures. The asparagus plants were fully established as a commercial crop and had been cultivated by a local grower (Bennekom, The Netherlands) since 2017. White asparagus was grown in drills under translucent plastic foil, while green asparagus plants were grown in the standard manner in an open field. The shoots were harvested in the 2019 and 2020 seasons.

In both seasons, harvesting of spears was done on a single day of a specific week and always around 10 a.m. In 2019, all white asparagus samples were collected on 2 May (calendar week 18); green spears of Gijnlim and Grolim were harvested on 22 May (calendar week 21), and green Avalim on 5 June (calendar week 23). In 2020, all varieties were harvested on 13 May (calendar week 20) and on 5 June (calendar week 23).

Nine asparagus spears were collected per variety and per time-point and were directly transferred on ice to the lab. These were carefully rinsed under cold water and separated into three pooled samples of three spears per variety. Each pooled sample was prepared as described in Appendix A. All pooled samples were subsequently ground in liquid nitrogen using a grinding mill (IKA^®^, Staufen, Germany), and the powders were stored at −80 °C until further analysis.

### 4.2. Chemicals and Analytical Standards

Calcium chloride (CaCl_2_) and Ethylenediaminetetraacetic acid (EDTA) were purchased from Sigma-Aldrich (Zwijndrecht, The Netherlands). Ultrapure water (Milli-Q^®^ Reference Ultrapure Water Purification System, Merck, Amsterdam, The Netherlands) was used for the preparation of the EDTA solution (stock solution of 90 mM). A mix of n-alkanes (C_6_–C_21_) was prepared. All alkanes were purchased from Sigma-Aldrich (Zwijndrecht, The Netherlands). The analytical standards used for metabolite identification had a purity between 96 and 99%. All standards were all purchased from Sigma-Aldrich (Zwijndrecht, The Netherlands) except for methanethiol, 1,3-diethylbenzene, styrene and 1,4-diethylebnzene which were purchased at Greyhound Chromatography (Merseyside, UK). All standards were dissolved in methanol (Biosolve BV, Valkenswaard, The Netherlands). Methanol, formic acid and acetonitrile (Sigma-Aldrich, Zwijndrecht, The Netherlands) were used for the extraction and analysis of the non-volatile semi-polar compounds.

### 4.3. Sample Preparation for Volatile Analysis

For each ground sample, 500 (±3) mg fresh weight was transferred to a glass 10 mL GC screw-cap vial, with an ND18 magnetic screw cap fitted with a 8 mm silicone/PFTE septum (BGB Analytik, Harderwijk, The Netherlands). Samples were kept in liquid nitrogen until 0.73 g CaCl_2_ dihydrate and 0.5 mL 90 mM EDTA-NaOH (pH 7.5) were added to give final concentrations of 5 M and 50 mM, respectively, to deactivate endogenous enzymes as previously described by Tikunov et al. (2005) [50]. Subsequently, samples were thoroughly vortexed until homogeneous. For the preparation of the QC samples, 500 mg of all ground green and white spears were mixed.

### 4.4. Extraction of Volatiles and Analysis with HS-SPME GC-MS

The trapping of volatiles by SPME and the GC-MS settings were essentially as described by Diez-Simon et al. (2020) [51] with certain modifications. The asparagus extracts were incubated at 50 °C for 15 min with agitation (300 rpm), and subsequently, volatiles in the headspace were extracted by inserting the SPME-fiber PDMS/DVB/CAR 50/30 μm diameter, 1 cm length, (Supelco, Bellefonte, PA, USA) for 15 min at 50 °C without agitation using an MPS-2 autosampler (Gerstel, Mülheim an der Ruhr, Germany). Analytes were transferred to an Agilent GC7890A gas chromatograph coupled to a 5975C quadrupole mass spectrometer by thermal desorption in a Gerstel CIS4 at 250 °C for 2 min in split-less mode under a constant helium flow of 1 mL/min. The column used was a Zebron ZB-5MSplus with dimensions 30 m × 0.25 mm i.d. × 1.00 μm film thickness (Phenomenex, Utrecht, The Netherlands). The GC oven temperature started at 45 °C for 2 min, then increased at a rate of 8 °C/min to 250 °C, and finally at a rate of 15 °C/min to 280 °C and maintained for 3 min. The column effluent was ionised by electron impact at 70 eV with a scan range of *m*/*z* 33–330. The MS interface temperature was set to 280 °C. A mix of n-alkanes (C_6_–C_21_) was analysed with the same method to calculate the retention indices (RIs).

### 4.5. Extraction of Semi-Polar Non-Volatiles and Analysis with LC-MS

The extraction and analysis of non-volatile semi-polar components were performed based on the method described previously by De Vos et al. (2007) [52]. An aliquot of 300 (±3) mg fresh weight from each ground sample was taken to extract the semi-polar compounds in 99% methanol containing 0.133% formic acid (FA) in a 3:1 ratio (mL methanol-FA: mg sample) (final concentration 75% methanol/0.1%FA), followed by sonication and centrifugation for 15 min each. Chromatographic separation was done using an Acquity UPLC module (Waters) using a reversed Luna C18 column with dimensions 2.0 × 150 mm and 3 µm (Phenomenex, Utrecht, The Netherlands), at 40 °C, using a linear gradient from 5 to 75% acetonitrile acidified with 0.1% FA at a flow rate of 0.19 mL/min in 45 min. The injection volume was 5 μL. Compounds eluting were detected using a photodiode array detector (Waters 2996) and subsequently with a high-resolution Orbitrap FTMS mass spectrometer (Thermo Fisher Scientific, Landsmeer, The Netherlands) in negative electrospray ionisation (ESI) mode (*m*/*z* 95–1350), using the mass calibration and other settings as recently described by van Treuren et al. (2018) [53]. Tandem MS (MS/MS) spectra were also generated on QC samples for manual identification of specific compounds.

### 4.6. Processing of Mass Spectrometry Data

Raw data were processed following an untargeted metabolomic workflow. Both GC-MS and LC-MS raw data were processed using the MetAlign software [54] for baseline correction, peak picking (S/N > 3) and alignment of the mass signals. Mass signals that were present in less than three samples were discarded, considering that three biological replicates were analysed per variety. Subsequently, mass features were reconstructed to potential compounds using the MSClust workflow [55].

Volatile metabolites were identified based on matching of the reconstructed mass spectra and calculated RIs with authentic reference standards and using the NIST17 Mass Spectral library and in-house databases. Non-volatile metabolites were putatively identified based on their molecular ion mass match with the online databases KnaPSAck [56], Dictionary of Natural Products [57] and mzCloud [58] within a mass deviation of 5 ppm. In addition, MS/MS fragmentation patterns of selected compounds were manually compared to those in literature. Having analysed in negative ESI mode, the main detected ions were [M−H]^−^, its formic acid adduct [M+FA−H]^−^, and for a few compounds also [2M−H]^−^. The in-source and MS/MS fragments for most compounds comprised a loss of hexose (neutral loss of 162.05282), deoxyhexose (neutral loss of 146.05791), pentose (neutral loss of 150.052823), a methyl group (neutral loss of 14), CO_2_ (neutral loss of 44), and the aglycone of the related metabolite. All these indicative masses were used, when possible, to verify the annotations of the identified compounds. The levels of identification given follow the Metabolomics Standards Initiative suggestions [24]; level 1 identified compounds have been verified with authentic reference standards, level 2 identified compounds are those with a high match of mass spectrum (and RI for VOCs); level 3 are compounds with low match of mass spectrum (or RI for VOCs); level 4 are compounds that the match of mass spectrum (and RI for VOCs) is too low to provide a putative annotation.

### 4.7. Statistical Analysis and Visualisation Tools

Prior to unsupervised multivariate statistical analysis, processed GC-MS and LC-MS data were log-transformed and autoscaled [59]. The LC-MS dataset from 2020 was further corrected for batch effects based on injection order followed by an additional step of peak filtering, using the R-code package BatchCorrMetabolomics 0.1.14 as described by Wehrens et al. (2016) [60]. The processed GC-MS and LC-MS data were subjected to principal components analysis (PCA) separately per year. Furthermore, Student’s *t*-tests were performed for comparing the profiles between green and white types or the two time-point measurements, and analysis of variance (ANOVA) with Tukey’s honestly significant difference (HSD) post hoc tests were performed for the comparisons between the three varieties per type. The packages used for these tests were sklearn 0.24.1 [61] and statsmodels 0.12.2 in Python version 3.9.2 (19 February 2021). The obtained *p* values were adjusted for false discovery rate (FDR) using the algorithm of Benjamini and Hochberg [62]. FDR adjusted *p* values equal or lower than 0.05 were considered significant. Visualisation of data was performed using RStudio with R version 4.0.3 (10 October 2020), Python version 3.9.2 (19 February 2021) and Microsoft Excel for Microsoft 365 (version 2104, 2021).

## 5. Conclusions

In conclusion, the contrasting metabolite profiles of green and white asparagus were observed and illustrated. The main differences were reproducible across two consecutive harvest years and can be linked to the cultivation method and the direct environment of the growing shoots. Certain previously reported key asparagus volatiles and non-volatiles were detected at similar levels in the two types and many new metabolites (both identified and unknowns) were also described. Such metabolites are proposed to contribute to the characteristic asparagus profile and may potentially influence sensory impact. Concerning the asparagus flavour profile, a considerable number of aroma compounds were found to be highly abundant in the upper parts of the shoots, suggesting these may have a richer flavour compared to the basal stem parts. These findings provide valuable insights into the metabolome of raw asparagus which can help to better understand crop chemistry. Finally, these results can contribute to reconsidering the applicability of discarded asparagus waste after harvesting as a valuable source of flavour compounds.

## Figures and Tables

**Figure 1 metabolites-11-00708-f001:**
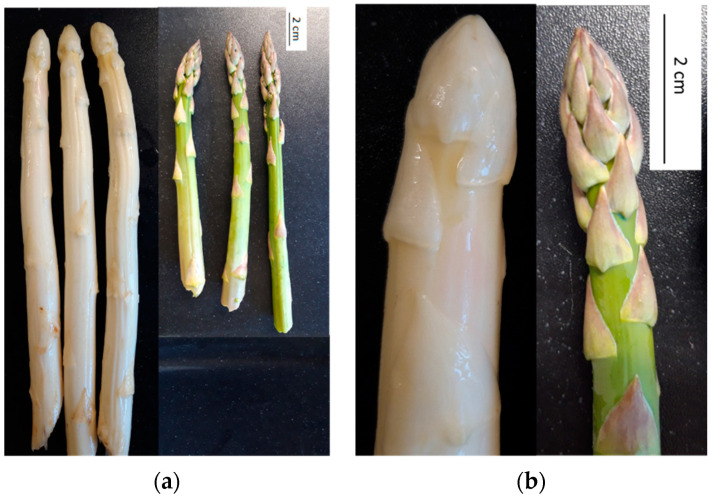
Clear differences in appearance are visible between white and green asparagus spears: (**a**) three freshly harvested white and green spears; (**b**) zoom in to reveal the cladodes and general tip morphology.

**Figure 2 metabolites-11-00708-f002:**
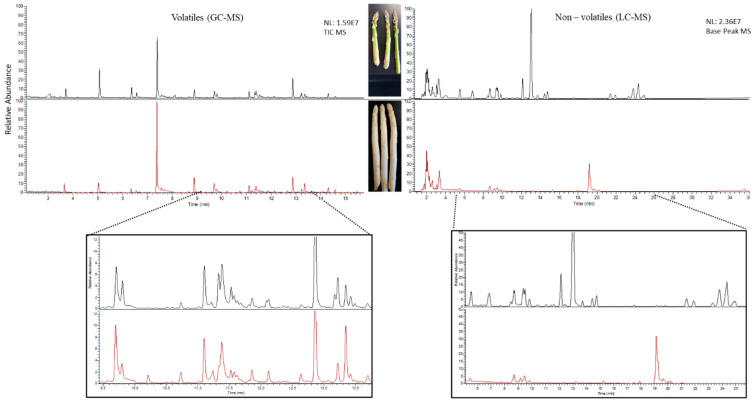
Raw chromatograms obtained from the GC-MS (**left**) and LC-MS (**right**) analysis of green (**upper**) and white (**lower**) asparagus spears.

**Figure 3 metabolites-11-00708-f003:**
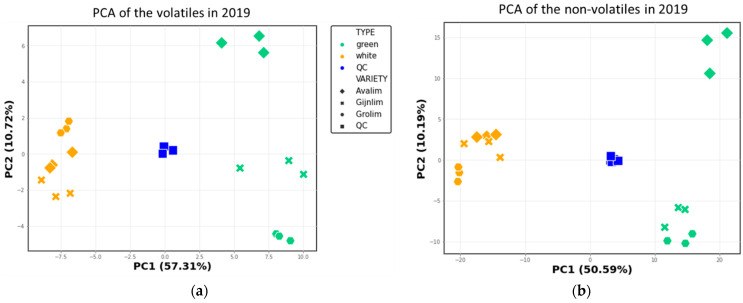
PCA score plots based on (**a**) 84 volatiles and (**b**) 442 non-volatiles detected in the green and white asparagus spears harvested in 2019: the first two PCs are presented, and the explained variance is shown in parentheses on the axes. Colours indicate the asparagus type; symbol shapes indicate asparagus variety.

**Figure 4 metabolites-11-00708-f004:**
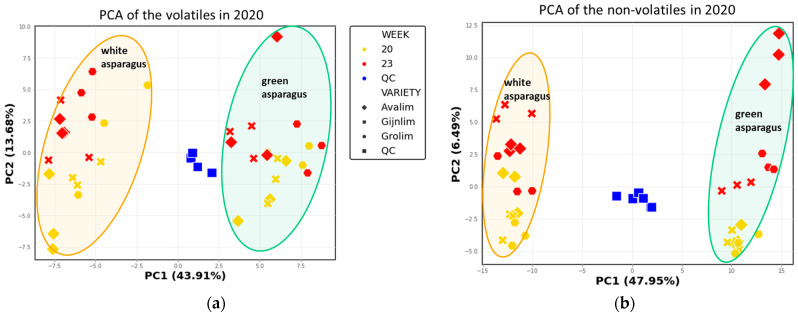
PCA score plots based on (**a**) 80 volatiles and (**b**) 255 non-volatiles detected in the green and white asparagus spears harvested in 2020: the first two PCs are given, and the explained variance is shown in parentheses on the plot axes. The ellipses indicate asparagus type; colour indicates the calendar harvest week; symbol shape indicates asparagus variety.

**Figure 5 metabolites-11-00708-f005:**
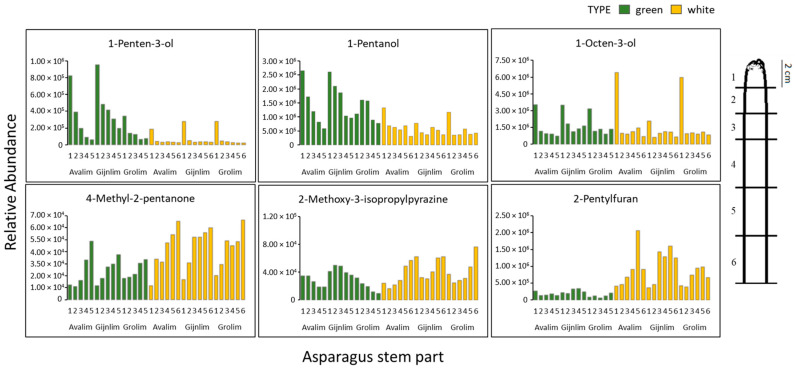
Relative abundance of selected identified volatiles in shoot sections of the asparagus varieties in their green (5 sections) and white (6 section) forms. Colours are based on the asparagus type. The numbers on the x axis correspond to the relevant spear sections as indicated on the right of this figure; each of the upper 3 sections represents 2 cm of spear, while the following 3 sections are each 4 cm in length. Green spears were shorter than the white as demonstrated in Figure 1, thus there were fewer green spear sections available for analysis.

**Table 1 metabolites-11-00708-t001:** Overview of characteristic (**a**) volatiles and (**b**) non-volatiles detected in either significantly different abundances between raw green and white asparagus (adjusted *p* value ≤ 0.05) or in only one asparagus type based on the two seasons. The CAS numbers (when applicable), experimental retention indices (RIs) and calculated monoisotopic mass (M) are given. The level of identification (LOI) provided is based on the Metabolomics Standards Initiative guidelines [24]. For a complete list of all detected compounds in both seasons (2019 and 2020) see Appendix A.

(a) Volatiles							
Metabolite Name	CAS	RI	LOI	Metabolite Name	CAS	RI	LOI
Significantly More Abundant in Green				Significantly More Abundant in White			
Methanethiol	74-93-1	548	1	Hexanal	66-25-1	794	1
Pentane	109-66-0	567	1	1,3-Dimethylbenzene	108-38-3	872	2
2-Methyl-3-buten-2-ol	115-18-4	623	1	2-Heptanone	110-43-0	890	2
2-Methylfuran	534-22-5	623	1	Hexanoic acid	142-62-1	964	3
1-Penten-3-ol	616-25-1	678	1	2-Pentylfuran	3777-69-3	993	1
3-Pentanone	96-22-0	689	2	unidentified	-	1050	4
Pentanal	110-62-3	692	1	1,3-Diethylbenzene	108-38-3	1056	1
1-Pentanol	71-41-0	758	1	(E)-2-Octenal	2548-87-0	1062	1
3-Methylbutanoic acid	503-74-2	824	3	1,2-Dimethoxybenzene	91-16-7	1146	1
(E)-2-Hexenal	6728-26-3	850	1	(E,E)-2,4-Nonadienal	5910-87-2	1223	2
3-Methylhexanal	19269-28-4	864	2	(E,E)-2,4-Decadienal	25152-84-5	1325	2
Heptanal	111-71-7	900	1	2,6-Di-tert-butylbenzoquinone	719-22-2	1477	3
Octanal	124-13-0	968	1				
1-Octen-3-one	4312-99-6	977	2				
6-Methyl-5-hepten-2-one	110-93-0	984	1				
**(b) Non-Volatiles**							
**Metabolite Name**	**CAS**	**M**	**LOI**	**Metabolite Name**	**CAS**	**M**	**LOI**
**Detected only in green**				**Detected only in white**			
Unidentified		356.1116	4	Furostane-3,22,26-triol		1052.541	2
3-O-Feruloylquinic acid	1899-29-2	368.1113	2	Unidentified		1050.525	4
Nicotiflorin	31921-42-3	594.1594	2	Spirostane-3,17-diol		756.4305	2
Kaempferol-sinapoyl-trihexose		978.2655	3	Shatavarin VI OR V		886.4996	3
Blumenol C glucoside	62512-23-6	372.2211	3	Spirost-5-ene-3,21-diol		884.4779	3
Ononin	486-62-4	430.1271	3				
**Significantly ore abundant in green**				**Significantly more abundant in white**			
Chlorogenic acid	906-33-2	354.0957	2	Unidentified		682.2542	4
3-p-Coumaroylquinic acid	87099-71-6	338.1008	2	Unidentified		265.9923	4
Asparagusic acid ester isomer II		312.0399	3	Unidentified		674.3497	4
Asparagusic acid ester isomer IV		312.0399	3	Protodioscin	55056-80-9	1048.547	2
Unidentified		432.2002	4	Shatavarin IX		902.4943	2
Unidentified		430.1903	4	Desglucomusennin		868.4888	3
Rutin	153-18-4	610.1543	2				
Isoquercetin	482-35-9	464.0964	2				
Unidentified		414.1321	4				
Unidentified		444.1428	4				
Unidentified		676.3738	4				
Quercetin-3-O-glucosylrutinoside		772.2074	2				

## Data Availability

The data presented in this study are available on request from the corresponding author. The data are not publicly available due to current project restrictions.

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
