# Peer review of "Metabolomics Reveals Heterogeneity in the Chemical Composition of Green and White Spears of Asparagus (A. officinalis)"

_metabolites, 2021, doi:10.3390/metabo11100708_

Round 1
Reviewer 1 Report
Overall thoughts on this study are that the experiments have been well performed using appropriate methods, however I am struggling to see the importance of this work and that is my biggest concern with this. My main thoughts on the manuscript itself is that it is far too long and all sections need to be significantly reduced in length I would strongly suggest by at least half. In its current form it contains information that is not relevant to the question that the study is trying to answer making it hard to follow what the authors are trying to say, for example this is a paper looking at crop chemistry the information and figure looking at the crop morphology is irrelevant to this.
The introduction needs to be shortened it is too long, it would strongly suggest shortening it (by about half) and it also needs to be given focus in its current state it is very hard to tell why the authors are doing this work they state “Such an investigation is needed to provide the fundamental chemical basis to understanding the physiology and chemistry of the crop, as well as to exploring the pathways involved in the formation of flavour precursors and to the vegetable flavour itself after food preparation.” However they make no effort to explain why knowing the difference between green and white asparagus is important this is important, what will this lead to next? What will this enable other researchers to do? They also need to be clear why it is important to look at the differences in the spatial distribution of metabolites, the authors state “It was thus hypothesized that these developmental and morphological differences will be linked to spatial phytochemical differences along the spear.” but if we eat the whole stem does it matter in which part of the stem that the compound came from.
The main finding of this paper is the metabolites found to differ between green and white varieties so this information needs to be more clearly displayed in the main paper, I would strongly suggest the inclusion of a main text table showing the measured metabolites and how the differ (or don’t) between the varieties.
One of the strengths of this study is the fact that they looked at the differences in two growing seasons effectively validating their findings a step that should be taken in more metabolomic studies. I feel that the results should be talked about more in terms of simply the differences between green Vs white based only on the metabolites that validate between the two growing seasons, this will help shorten the paper and make the results easier to follow for the reader.
The method that has been used is well described and from this I am confident that the platforms used are both powerful and had appropriate measures for assuring data quality. I would suggest shortening this section.
I feel that the figures used in this study are not particularly informative, for example the PCA plots in figure 3 and 4 show that there are large differences in composition between white and green varieties and smaller differences between cultivars, however given the morphological differences especially between white and green this will already be known by the reader. Also figures A3 and A4 I don’t feel add any more information than is stated in the text.
Author Response
Dear Reviewer,
Thank you very much for your valuable feedback and comments. Please find enclosed our actions given your feedback (see the attachment).

Reviewer 2 Report
The study comprehensively compared the temporal and spatial metabolomes of raw white and green asparagus spears using combined HS-SPEM GC-MS and UPLC-MS approaches in three different asparagus cultivars grown at the same location for two seasons. This research has updated understanding of green and white asparagus chemistry, linking metabolome to asparagus flavor and sensory quality.
Only three minor issues:
- There are many terms to describe the two asparagus spears, for example, two types, two cultivars, two forms, to crop types, two crops, two spear types…. It is better to use the same term throughout the MS.
- 2) In M&M section (line 522-523), why did authors put harvested spears on ice. It is a cold stress, which will induce differential metabolomic changes between green and white spears, because they are supposed to response to cold differently.
- 3) Since both Avalim and Gijnlim are considered suitable for both green and white asparagus production, while Grolim is less suited as a green crop, it is curious to know the different metabolomic changes in these two cultivars.
Author Response

(The authors gave the same response as above.)

Round 2
Reviewer 2 Report
The authors have adderessed properly all my concerns and significantly improved the MS based on the comments from another reviewer.
I have no more concerns.